# The Mechanism of Dynamic Interaction between Doxorubicin and Calf Thymus DNA at the Single-Molecule Level Based on Confocal Raman Spectroscopy

**DOI:** 10.3390/mi13060940

**Published:** 2022-06-13

**Authors:** Ruihong Zhang, Jie Zhu, Dan Sun, Jie Li, Lina Yao, Shuangshuang Meng, Yan Li, Yang Dang, Kaige Wang

**Affiliations:** 1State Key Laboratory of Cultivation Base for Photoelectric Technology and Functional Materials; National Center for International Research of Photoelectric Technology & Nano-Functional Materials and Application; Shaanxi Provincial Key Laboratory of Photoelectric Technology; Institute of Photonics and Photon-Technology, Northwest University, Xi’an 710069, China; 14760546324@163.com (R.Z.); ozhujieo@163.com (J.Z.); sund@nwu.edu.cn (D.S.); lj15891397079@163.com (J.L.); lemonade@163.com (L.Y.); mengss1129@163.com (S.M.); yangdang202011@163.com (Y.D.); 2School of Science, Xi’an Shiyou University, Xi’an 710069, China; liyan67@xsyu.edu.cn

**Keywords:** doxorubicin, calf thymus DNA, binding site, dynamic process, Confocal Raman spectroscopy

## Abstract

It is of great fundamental significance and practical application to understand the binding sites and dynamic process of the interaction between doxorubicin (DOX) and DNA molecules. Based on the Confocal Raman spectroscopy, the interaction between DOX and calf thymus DNA has been systemically investigated, and some meaningful findings have been found. DOX molecules can not only interact with all four bases of DNA molecules, i.e., adenine, thymine, cytosine, guanine, and phosphate, but also affect the DNA conformation. Meanwhile, the binding site of DOX and its derivatives such as daunorubicin and epirubicin is certain. Furthermore, the interaction between DOX and DNA molecules is a dynamic process since the intensities of each characteristic peaks of the base, e.g., adenine, cytosine, and phosphate, are all regularly changed with the interaction time. Finally, a dynamic mechanism model of the interaction between DOX and DNA molecules is proposed; that is, there are two kinds of interaction between DOX and DNA molecules: DOX-DNA acts to form a complex, and DOX-DOX acts to form a multimer. The two effects are competitive, as the former compresses DNA molecules, and the latter decompresses these DNA molecules. This work is helpful for accurately understanding and developing new drugs and pathways to improve and treat DOX-induced cytotoxicity and cardiotoxicity.

## 1. Introduction

The research on the therapeutic mechanism and developments of anticancer drugs has always been a frontier hot topic. Many drugs for the prevention and treatment of cancer are designed with DNA molecules as the target [1,2,3]. At present, it is generally believed that the interaction between drug molecules and DNA molecules is mostly non-covalent bonding force, including electrostatic force, groove binding, and intercalation. At the single molecular level, understanding the action mode of anticancer drugs and DNA molecules and clarifying the action mechanism of anticancer drugs have scientific and practical significance for evaluating medicine efficacy and guiding drug development.

Doxorubicin (DOX) is a highly efficient and broad-spectrum anthraquinone anticancer drug which can be used to treat various diseases, such as leukemia, lymphoma, breast cancer, liver cancer, etc. [4,5,6]. However, cytotoxicity and cardiotoxicity limited its clinical application; it was found that DOX could cause cardiomyocyte damage and severe cardiotoxicity which led to arrhythmias and even fatal heart failure in patients [7,8,9,10]. In recent decades, researchers have been investigating natural anthracycline compounds with low toxicity and high activity, and have developed a number of semi-synthetic analogues [11,12,13]. Unfortunately, medicine with clinical advantages similar to DOX has not been created; the main reason is that the anticancer mechanism of DOX is not still wholly understood, thus the efficacious derivatives cannot be designed and carried out.

At present, the viewpoints on the mechanism of DOX mainly include the following. One opinion is that the DOX generates semiquinone free radical through reduction reaction under the action of oxidoreductases, and then the reoxidation of semiquinone free radical initiates the generation of reactive oxygen species (ROS), which causes oxidative stress, DNA damage, peroxidation membrane damage, and ultimately results in the death of cancer cells [14,15]. Another opinion is that DOX directly intercalates into DNA molecules, thereby breaking the complex between DNA and Topo II enzyme, which prevents the reclosure of the DNA double helix structure of cancer cells, and eventually leads to the death of cancer cells [16,17,18]. However, the results of research on the insertion sites on DNA molecules are still inconsistent. Some results show that DOX intercalates into the base pair of cytosine and guanine of DNA, while other data indicate that DOX intercalates into the base pair of adenine and thymine. For example, Zhang et al. found that the DOX intercalates into the base pair of cytosine and guanine investigated with the fluorescence correlation spectroscopy [19]; Lee et al. found that DOX intercalates into the base pair of adenine and guanine by the surface-enhanced Raman scattering and the UV-resonance Raman Spectroscopy [20]; Wang et al. suggested that DOX electrostatically interacts with the phosphate group of DNA molecules investigated by the infrared spectroscopy [21]; while Cristina et al. found that DOX can bind with both base pairs of A-T and G-C by comparing the equilibrium constants [22]. More unfavorably, the aforementioned studies only focused on the static process of the interaction between DOX and DNA molecules, while the process of the interaction between DOX and DNA molecules was rarely reported [23]. It is a dynamic process that the action of drugs, e.g., the interaction between DOX and DNA molecules. Both the binding sites of the DOX and DNA molecules and the dynamic interaction mechanism are fundamental problems that need to be investigated with great effort.

In general, there are effective methods to study the interaction between DOX and DNA, e.g., Raman spectroscopy [24], fluorescence correlation spectroscopy [19], infrared spectroscopy [21], the electrochemical methods [22], etc. The Confocal Raman spectroscopy is an optical detection technology that combines the Raman spectroscopy and the confocal technology. It can effectively eliminate the signal interference outside the focal plane, and its properties such as the spatial resolution, signal-to-noise ratio, and accuracy are all higher than that of the ordinary Raman spectroscopy [25]. The position, intensity, and line width of Raman spectra can provide information on the vibration and rotation of sample molecules; Confocal Raman spectroscopy technology can realize the “fingerprint identification” of the changes of certain chemical bonds and functional groups in the biomacromolecules caused by the interaction between drugs and biomolecules [26,27,28,29].

In the following, Confocal Raman spectroscopy was used to investigate the binding sites and the dynamic interaction process between DOX and calf thymus DNA (ctDNA) molecules. Then, the interacting sites of daunorubicin (DAU) and epirubicin (EPI) with ctDNA and their dynamic process were also analyzed as experimental validation.

## 2. Materials and Methods

The samples such as DOX, DAU, and ctDNA were purchased from Sigma-Aldrich Corporation. EPI was purchased from Shanghai Jinsui Biotechnology Co., Ltd. The whole experimental solution was prepared with deionized water, in which the concentrations of DOX, DAU, and EPI were $5\times 10^{-2}\;M$, and the concentration of ctDNA was $9.8\times 10^{-2}\;M$. During experiments, an appropriate amount of DOX, DAU, EPI, and ctDNA solutions were measured respectively, and then put them into three different reagent bottles that contained the same amount of DNA solution by a pipette gun. Subsequently, they were mixed them thoroughly and incubated at 4 °C for a period time; as a result, the DOX-DNA, DAU-DNA, and EPI-DNA solutions were obtained.

Raman spectra were acquired by an Alpha 500R Confocal Raman microscopy system (Alpha 500R, WITec GmbH, Ulm, Germany). Briefly, a fiber-coupled 785 nm semiconductor laser was collimated into a 20× objective lens (NA = 0.2, N-A chroplan, Zeiss, Oberkochen, Germany) for Raman excitation and spectral measurements. During the experiment, the power of the laser was 10 mW, the spot diameter of 20× objective was 1 μm, and the power density was 1.3 × 10^7^ mW/mm^2^. The spectral signal was recorded by a spectrometer (UHTS300, WITec GmbH, Germany) incorporated with a 600 ${\mathrm{gmm}}^{-1}$ grating blazed at 500 nm, with a back-illuminated deep-depletion charge-coupled device (CCD, Du401A-BR-DD-352, Andor Technology, Belfast, UK) camera working at −60 °C. When measuring Raman spectra, each spectrum was measured three times and the acquisition time was approximately 20 s.

Before observing and recording the Raman spectral signals, a 10 μL droplet of solution (ctDNA, DOX, DAU, EPI, DOX-DNA, DAU-DNA, and EPI-DNA) was dripped onto the gold-plated substrate and dried at room temperature. The spectral data were firstly preprocessed with the data analysis software of WITec Project 4, including removing cosmic rays from the original spectrum. All Raman spectra were clipped to the fingerprint region (400–1800 ${\mathrm{cm}}^{-1}$), and the background interference was removed by 9-order polynomial fitting. The spectrum was smoothed using 5-order Savitzky-Golay (SG).

Furthermore, the absorption spectra of the DOX-DNA compounded solution were detected by the ultraviolet-visible (UV-vis) spectroscopy (DS-11). The light source of UV-vis spectrophotometer is long-life pulsed Xenon lamp, whose detector is a silicified CCD array with 2048 elements, and the detection accuracy and resolution reach to 1 nm.

## 3. Results and Discussion

### 3.1. Raman Spectra of ctDNA Molecule

Figure 1a shows the diagrammatic sketch of ctDNA molecular structure; Figure 1b shows the average Raman spectra of ctDNA fiber (blue line) and ctDNA solution (black line) with a concentration of $9.8\times 10^{-2}\;M$.

From Figure 1b, it can be observed clearly that the main characteristic Raman peaks of ctDNA are at 733, 785, 877, 1006, 1100, 1378, 1484, 1574, 1624, and 1660 ${\;\mathrm{cm}}^{-1}$. These characteristic peaks reveal the structural information of ctDNA. The bands at 733, 785, 1378, 1484, 1574, and 1660 ${\mathrm{cm}}^{-1}$ are the characteristic peaks of base pairs, 1006 ${\mathrm{cm}}^{-1}$ is the characteristic peak of deoxyribose, and 1100 ${\mathrm{cm}}^{-1}$ is the characteristic peak of the phosphate group.

Table 1 lists the detailed attribution information of the main Raman peaks of ctDNA [20,30,31,32,33,34,35,36]. The 733${\;\mathrm{cm}}^{-1}$ is assigned to the symmetric stretching vibration of adenine (A). The 785 ${\;\mathrm{cm}}^{-1}$ is mainly related to the symmetric stretching vibration of cytosine (C). The 877 ${\mathrm{cm}}^{-1}$ is related to the C of DNA conformation. The 1006$\;{\mathrm{cm}}^{-1}$ is the contribution of C-O stretching vibration of deoxyribose. The 1100${\;\mathrm{cm}}^{-1}$ is attributed to symmetric stretching vibration of O=P=O in ${\;\mathrm{PO}}_{2}{}^{-}$. The 1378${\;\mathrm{cm}}^{-1}$ is attributed to the thymine (T). The 1484$\;{\mathrm{cm}}^{-1}$ is attributed to the guanine (G). The 1574 ${\mathrm{cm}}^{-1}$ is attributed to the adenine (A). The 1624 ${\mathrm{cm}}^{-1}$ is attributed to the N7C5 + C8N7 vibration of A. The 1660$\;{\mathrm{cm}}^{-1}$ is the contribution of C2=O stretching vibration of C.

### 3.2. Raman Spectra of DOX

Figure 2a shows the molecular structure of DOX; Figure 2b shows the average characteristic Raman spectra of DOX solution with a concentration of $5\times 10^{-2}\;M$.

From Figure 2b, it can be seen clearly that the characteristic Raman peaks of DOX are mainly at 440, 462, 989, 1082, 1209, 1241, 1299, 1448, 1578, and 1637 ${\mathrm{cm}}^{-1}$. These characteristic peaks reveal abundant structural information of DOX molecules [37,38]. The bands at 440 and 462${\;\mathrm{cm}}^{-1}$ are assigned to C-C-O and C-O respectively. The band at 989 ${\;\mathrm{cm}}^{-1}$ is attributed to the vibration of the C-C of ring A (as shown in Figure 2a). The bands at 1082, 1209, and 1241 ${\;\mathrm{cm}}^{-1}$ relate to C-O, C-O-H, and C-H, respectively. The 1578${\;\mathrm{cm}}^{-1}$ is attributed to ring stretching and the band at 1637${\;\mathrm{cm}}^{-1}$ is attributed to C=O. Generally, these characteristic peaks (440, 462, 989, 1082, 1209, 1241, 1578, and 1637 cm^−1^) are the crucial criteria to identify whether the solute is DOX (DAU/EPI), while 1299 cm^−1^ and 1448 cm^−1^ are not the crucial criteria. The band at 1299 cm^−1^ is assigned to the vibration involving in-plane C-O, C-O-H, and C-H bending modes; the band at 1448 cm^−1^ is assigned to the C=C and the C-C stretching of the aromatic hydrocarbons.

### 3.3. Raman Spectra of DOX-DNA Complex

Figure 3a shows average Raman spectra of ctDNA solution (blue line); Figure 3b shows the average Raman spectra of DOX-DNA complex (black line) obtained by DOX interacting with ctDNA for 24 h.

Column 2 in Table 1 lists the detailed attribution information of Raman spectral characteristic peaks of DOX-DNA. Comparing the characteristic peaks of ctDNA with DOX-DNA complex, some distinguishing features can be found, e.g., the characteristic peak at 877${\;\mathrm{cm}}^{-1}$ in the Raman spectrum of the original ctDNA solution, indicating that the DNA may be C-DNA under this experimental condition [30,31,32,33]. When DOX interacts with DNA for 24 h, Raman peaks at 841 and 857${\;\mathrm{cm}}^{-1}$ appear, indicating that DNA may coexist with B-DNA and C-DNA after DOX interaction with DNA. The Raman spectrum of the DOX-DNA complex has both redshift and blueshift. The redshift may be due to the influence of the hydrogen bond [34]. For example, the characteristic peak at 1100${\;\mathrm{cm}}^{-1}$ belongs to the symmetrical stretching vibration of phosphate ion ${\mathrm{PO}}_{2}^{-}$, and the Raman peak at 1100 ${\mathrm{cm}}^{-1}$ moves to 1089${\;\mathrm{cm}}^{-1}$ due to the hydrogen bond between phosphate ion ${\mathrm{PO}}_{2}^{-}$ of DNA molecule with ${\mathrm{NH}}_{2}$ and OH of DOX. The characteristic peak at 1660${\;\mathrm{cm}}^{-1}$ belongs to the C2=O stretching vibration of cytosine. The redshift from 1660 to 1657${\;\mathrm{cm}}^{-1}$ is caused by the deformation of the hydrogen bond between C2=O of cytosine and ${\mathrm{NH}}_{2}$ of DOX. Similarly, the characteristic peak at 1624${\;\mathrm{cm}}^{-1}$ is mainly attributed to the vibration of N7C5 + C8N7, and the redshift from 1624 to 1600${\;\mathrm{cm}}^{-1}$ may also be due to the formation of the hydrogen bond between N7 of A and DOX. Due to the influence of DOX, both bands at 1484${\;\mathrm{cm}}^{-1}\;$(attribute to G) and 1378${\;\mathrm{cm}}^{-1}\;$(attribute to T) have blueshifts. Evidently, the DOX not only interacts with adenine, thymine, cytosine, guanine, and phosphate ion ${\mathrm{PO}}_{2}^{-}$, but may also affect the DNA conformation.

### 3.4. Raman Spectra of DAU, EPI, DAU-DNA, and EPI-DNA

In order to further confirm the binding sites of DOX with DNA, we also analyzed Raman spectra of DOX derivatives, i.e., DAU and EPI, and their complexes with DNA.

Figure 4 shows the chemical structure diagram of (a) DAU and (b) EPI, respectively. The red circle in the figure indicates their structural differences from DOX. Comparing Figure 4 to Figure 2a, it is found that the structures of the three anticancer drugs are very similar, but they also have differences. For example, the hydroxyl at position 14 of DOX is changed into hydrogen in DAU, and the hydroxyl at amino sugar part of EPI is changed from cis to trans.

Figure 5a–c are the Raman spectra of DOX, DAU and EPI, respectively. In Figure 5, the black curve represents the average Raman spectrum of DOX solution; the red curve represents the average Raman spectrum of DAU solution; the blue curve represents the average Raman spectrum of EPI solution.

Because the DOX, DAU, and EPI have similar structures, here DAU and EPI were mainly used to verify the binding sites between DOX and DNA; that is, the adenine, thymine, cytosine, guanine, and phosphate. In the process of normalizing the ordinate of the Raman spectrum as shown in Figure 5, the characteristic peaks corresponding to the maximum value of all spectra are consistent, and the different spectra are comparable.

Due to their similar chemical structures and Raman spectra [39], the main characteristic Raman peaks such as 440, 462, 1082, 1209, and 1241 ${\;\mathrm{cm}}^{-1}$ are found in DOX, DAU, and EPI solutions. Raman peaks at 440 and 462 ${\mathrm{cm}}^{-1}$ belong to C-C-O and C-O, respectively. The bands at 1082, 1209, and 1241 ${\mathrm{cm}}^{-1}$ respectively relate to C-O, C-O-H, and C-H. The Raman spectra of DOX, DAU, and EPI showed strong similarity, with all major characteristic peaks being the same, and only a small difference was found at 1300 cm^−1^. This characteristic peak was more significant in DOX and EPI, but less significant in DAU.

It should be noted that Figure 5 was only used to illustrate the structural similarity of DOX, DAU, and EPI. Their differences are not the focus of this study. Therefore, there are no in-depth analyses on the Raman spectra of the different drugs in Figure 5.

In Figure 6, the black curve represents the average Raman spectrum of ctDNA solution; the red curve represents the average Raman spectrum of DOX-DNA complex obtained by DOX interaction with DNA for 24 h; the blue curve represents the average Raman spectrum of DAU-DNA complex obtained by DAU interaction with DNA for 24 h; the green curve represents the average Raman spectrum of EPI-DNA complex obtained by EPI interaction with DNA for 24 h.

By comparing the average Raman spectra of DOX, DAU, and EPI interacting with DNA, it is found that there are also the same peak shifts at 1100, 1378, 1484, and 1660 ${\mathrm{cm}}^{-1}$. The only difference is the Raman peak at 1624${\;\mathrm{cm}}^{-1}$. The Raman peak at 1624 ${\;\mathrm{cm}}^{-1\;}$ will be shifted to about 1600 ${\;\mathrm{cm}}^{-1}$ by DOX and EPI interacting with DNA, but the Raman peak shift at 1624${\;\mathrm{cm}}^{-1}$ by DAU interacting with DNA is very small. The main reason may be attributed to that both DOX and EPI are hydroxyl at position 14, while DAU is hydrogen. The interaction sites of DOX and DNA are shown in Figure 7.

### 3.5. Raman Spectra of Dynamic Process of DOX Interaction with ctDNA

In order to understand and analyze the dynamic process of the interaction between DOX and DNA, an appropriate amount of DOX with DNA solution is mixed and incubated, and the characteristic Raman spectra are detected after a series of periods. Figure 8 displays the Raman spectra of DOX interaction with DNA for various times (0, 1, 2, 5, 7, 10, 24, 48, and 72 h).

From Figure 8, it can be obviously found that the intensities and peak positions of Raman spectra changed significantly at 726, 785, 1089, 1450, 1578, and 1657${\;\mathrm{cm}}^{-1}$. These characteristic peaks do reveal the structural information of DNA molecules [30,31,32,33,34,35,36]. Among them, the spectral peak at 726${\;\mathrm{cm}}^{-1}$ belongs to A, the characteristic peak at 785 ${\mathrm{cm}}^{-1}$ mainly belongs to C, the characteristic peak at 1089${\;\mathrm{cm}}^{-1}$ belongs to the symmetrical stretching vibration of phosphate ion ${\;\mathrm{PO}}_{2}^{-}$, the band at 1450${\;\mathrm{cm}}^{-1}$ belongs to the deoxyribose, the spectral peak at 1578${\;\mathrm{cm}}^{-1}$ mainly belongs to A, and the characteristic peak at 1657${\;\mathrm{cm}}^{-1}$ mainly belongs to C.

In order to clearly express the changing trend of Raman intensity with incubation time, Figure 9 shows the line charts of Raman intensities with incubation time at 726, 785, 1089, 1450, 1578, and 1657${\;\mathrm{cm}}^{-1}$, in which the horizontal axis is the logarithmic time and the vertical axis represents the Raman intensity.

From Figure 9, it can be obviously found that the intensity of Raman peaks at 726, 785, 1089, 1450, 1578, and 1657${\;\mathrm{cm}}^{-1}$ and has periodic increases and decreases with the increase of incubation time. The detailed changes of Raman intensity can be simply described as the following: the intensities of these Raman peak values decrease in 0–1 h, increase in 1–2 h, decrease again in 2–5 h, and increase again in 5–7 h. However, the degree of increase or decrease each time is different for different characteristic peaks. After approximately 10 h, the change trends are no longer regular. It should be point out that in order to ensure the reliability of the measurement data, the experiment was repeated 6 times under each different working condition, and then the average value was taken as the final result. 

The reason for the periodic decrease of Raman intensity may be attributed to DOX acting on adenine, cytosine, phosphate, and deoxyribose of DNA molecules, and reducing the contents of adenine, cytosine, phosphate, and deoxyribose, while the reason for the periodic increase of Raman intensity may be complicated. 

The increase of Raman spectral intensity reflects the increase in the number of counterparts in the mixing solution. Without increasing the number of DNA molecules, the main reason for this result may be that the DNA molecule combined with DOX exposes some bases such as adenine and cytosine, phosphate and deoxyribose, where the characteristic Raman peaks can be detected. In other words, the DOX molecule that has been bound to DNA molecule and formed a DOX-DNA complex may detach from the DNA molecule; therefore, those original acting sites, e.g., the bases, phosphate, or deoxyribose, are uncovered and will be detected and contribute to the Raman intensity value. The key question in this process is how DOX molecule leaves the DOX-DNA complex. Some research groups have reported that DOX molecules can combine with DOX molecules to form a multimer, such as dimer, trimer, or tetramer, when the number of DOX molecules is relatively large; for instance, C_DOX_/C_DNA_ > 0.3 [19,22,40]. C_DOX_ and C_DNA_ is the concentration of DOX and DNA molecules, respectively. In this way, it is possible that the DOX molecule on the DOX-DNA complex combines with free DOX molecules to form a multimer, and then leaves the DNA molecule.

### 3.6. Absorption Spectra of the Culture Medium of DOX and DNA Molecules

In order to determine the changes in the amount of DNA molecules in the solution after culturing DOX and ctDNA for a period of time, the absorption spectra of DOX-DNA culture medium at different times are also systematically measured.

UV-vis absorption spectrum can be used to analyze, determine, and infer the composition, content, and structure of substances of interest. The shape of UV-vis absorption spectrum curve and its maximum absorption wavelength reflect the composition of the substance, and the absorbance reflects the content of the substance.

Figure 10a shows the absorption spectrum of DNA molecules when DOX and ctDNA are mixed and cultured for various times, e.g., 0, 1, 2, 5, 7, 10, 24, 48, and 72 h. Figure 10b is the absorbance value of mixed solution of DOX-DNA around 258 nm at different incubating times, in which the horizontal axis is the logarithmic time and the vertical axis represents the Raman intensity.

From Figure 10, it can be clearly found that the intensity of the absorption spectrum of the DOX-DNA complex at 258 nm changes regularly with the increase of administration time. The change in the absorbance of DOX-DNA at 258 nm can be simply described as: the absorption peak decreases within 0–1 h, then increases within 1–2 h, and then decreases again within 2–5 h; however, the decrease is smaller. Afterwards, it gradually increases until 24 h; from 24 h to 48 h, the change is not obvious.

The decrease in absorbance intensity reflects a decrease in the amount and area of detectable ctDNA in the mixed solution, while the increase in absorption intensity indicates the increased number and area of DNA molecules exposed. Therefore, it is reasonable to speculate that during the period of experiments, the interaction between DOX and DNA brings about the change of the state and configuration of DNA constantly.

### 3.7. Molecular Mechanism of Interaction between DOX and DNA

Figure 11 is a schematic diagram of the dynamic process during the interaction between DOX and DNA, wherein (a–e) represent the changes about DNA morphology at different times after DOX administration, corresponding to the spectra at different times of 0, 1, 2, 5, and 7 h, as shown in Figure 9.

It can be seen from Figure 11, after the DNA molecule interacts with DOX, that its configuration exhibited a change process, namely, compaction, decompaction, recompraction, and redecompaction [41]. Meanwhile, the inset in the middle is an example showing the competitive process through which two DOX molecules attract the DOX molecule in the DOX-DNA complex to form a trimer (dimers or tetramers might also be formed with different amounts of DOX molecules) and leaving the complex. That is, if new DOX molecules continuously entered the solution or the concentration of DOX was greater than a certain value, during the process of interaction between DOX and DNA molecules, accompanied by the interaction among DOX and DOX molecules, the two interactions between DOX and DOX molecules or DOX and DNA molecules were in competition, and the interaction between DOX and DNA molecules was constantly changing and unstable. The competition between the two mechanisms made the interaction between DOX and DNA molecules more complicated.

Concretely speaking, the dynamic process of the interaction between DOX and DNA molecules can be described as the following: as shown in Figure 11a, the DNA molecules are in a free self-coiled state in solution before DOX molecules were added into the solution. When DOX molecules are dropped into the DNA solution, DOX molecules act on adenine, cytosine, phosphate radicals, and/or deoxyribose of DNA molecules, and result in the ctDNA strands bending, shrinking, and compressing, as shown in Figure 11b. After a period of action of approximately 2 h, as shown in Figure 11c, compared to the state of the DNA molecule shown in Figure 11b the DNA molecule is in a relatively relaxed state again, i.e., decompaction. At this time, the concentration of DOX is relatively high (C_DOX_/C_DNA_ > 0.3), and those free DOX molecules in solution can combine with the DOX which is in the DNA-DOX complex to form a multimer, e.g., dimer, trimer, and/or tetramer, and then these multimers detach from the DNA molecule, thus causing the DNA molecule to be stretched or loosened, i.e., in a decompressed state. It should be pointed out that the state and conformation of the DNA molecule here are not exactly the same as that of the DNA molecule shown in Figure 10a. As shown in Figure 11d, after administration of approximately 5 h, those free DOX molecules act on adenine, cytosine, phosphate, and/or deoxyribose of DNA molecules, and cause the decompacted DNA strands to be bent or contracted until compressed again, i.e., recompaction. It is also worth emphasizing that the state and conformation of the DNA molecule at this time are not exactly the same as that of the DNA molecule shown in Figure 11b; the sites of interaction between DOX and DNA molecules may not be the same. Subsequently, free DOX molecules in the solution can form a multimer with the DOX in the DOX-DNA complex, and then the multimer drags away the DOX molecule from the complex and meanwhile a possible action site on the DNA chain is exposed. The separation of the DOX molecule from the complex brings about the recompacted DNA molecule to stretch and relax again, and be in a decompressed state again, i.e., redecompaction, as shown in Figure 11e. For a similar reason, the DNA state and configuration are different from that shown in Figure 11a,c.

The inset surrounded by dashed lines in the middle mainly shows the detailed process of multimer formation and leaving the DOX-DNA complex, which can be divided into three main steps: (1) multiple free DOX molecules in the solution are moving close to the DOX molecule that has been bound to the DNA strand. (2) The DOX trimer is formed with two free DOX and the DOX of the complex, and then the trimer breaks away from the DNA chain; the trimer is only an example. (3) The multimer leaves and moves into the solution and the binding site on the DNA molecule is exposed, and new free DOX molecules will draw near and bind to the DNA molecule again.

It can be found that some phenomena in the experiments have been explained more credibly and reasonably by the mechanism model of interactions between DOX and DNA illustrated in Figure 11, which we propose based on our experimental results and those of other peer research groups [19,22,40]. We are conducting further investigations on this model, e.g., by using Dynamic Light Scattering (DLS), Atomic Force Microscopy (AFM), and so on, to determine the magnitude and type of force at the action site, the influence of DOX on the size, and morphology of DNA molecules.

## 4. Conclusions

Based on Confocal Raman spectroscopy, the binding sites of DNA molecules and DOX molecules and the dynamic process of their interaction were systematically studied, and some meaningful findings were found:

Firstly, the characteristic peaks of the adenine, thymine, cytosine, guanine, and phosphate in the DOX-DNA complex have redshift or blueshift, which means that all four bases A, T, C, G, and ${\mathrm{PO}}_{2}^{-}$ of DNA molecules can become the action site when DOX interacts with the DNA molecule and the DNA state and conformation can also be affected by the DOX molecule. Then, combined with the Raman spectral characteristics of DOX derivatives such as DAU and EPI, and the complex DAU-DNA, EPI-DNA, the binding site of DOX was confirmed when it interacts with ctDNA molecule. Moreover, the characteristic Raman spectra of DOX-DNA complexes in different periods were carefully compared and analyzed, and the intensities of each characteristic peaks representing adenine, cytosine, and phosphate were all regularly changed with the interaction time, which indicated that the interaction between DOX and DNA was a dynamic process. Finally, the dynamic mechanism of the interaction between DOX and DNA molecules is proposed as the following: some free DOX molecules bind with four bases and/or phosphates of DNA molecules, resulting in a DNA molecule compression; then, new free DOX molecules bind with the DOX molecule of the DOX-DNA complex and forms a multimer; subsequently, the multimer drags away the DOX and leaves the DNA molecule, thus the DNA molecule loosens and exposes a binding site (some previously shielded binding sites may also be exposed because the state and configuration of DNA molecules in solution are constantly changing), decompressing the DNA molecule; then, some new free DOX molecules bind to DNA molecule, causing the DNA molecule compression again, i.e., recompaction; and afterwards, some new free DOX molecules bind to the DOX molecule in the DOX-DNA complex and form a new multimer and leave, bringing about the DNA molecule decompression again, i.e., redecompaction; and so on.

To investigate and elucidate the binding sites and dynamic process of the interaction between DOX and DNA molecules, it is not only of fundamental significance for molecular pharmacology, but also has practical application for understanding the biological processes at a single molecule level, for the design of hybrid nanomaterials for medicine delivery based on DNA architecture, and for the development of new drugs and pathways to improve and perfect DOX cytotoxicity and cardiotoxicity.

## Figures and Tables

**Figure 1 micromachines-13-00940-f001:**
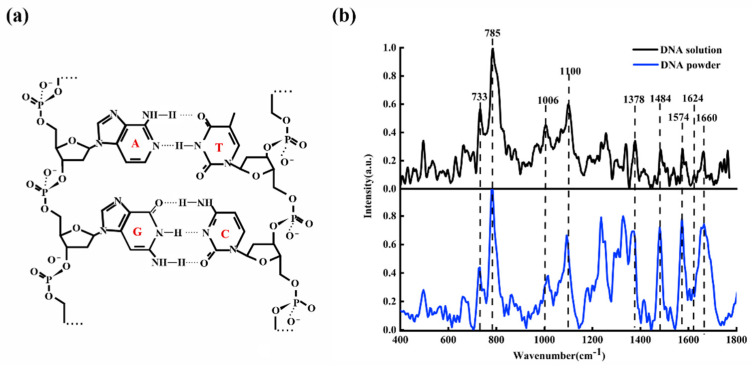
(**a**) The molecular structure of ctDNA; (**b**) Raman spectra of ctDNA fiber (blue line) and ctDNA solution ($9.8\times 10^{-2}\;M$, black line).

**Figure 2 micromachines-13-00940-f002:**
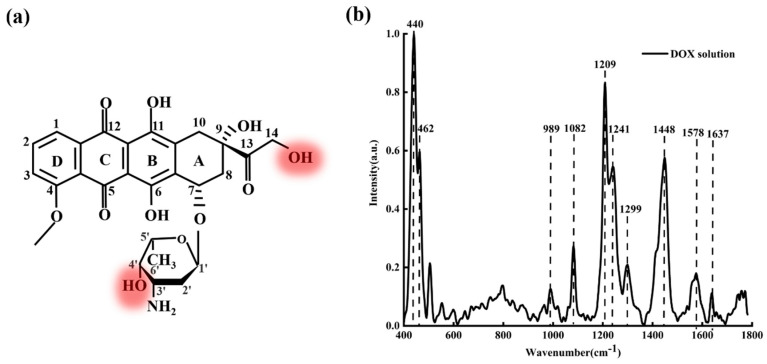
(**a**) The molecular structure of DOX and (**b**) Raman spectra of DOX solution ($5\times 10^{-2}\;M$).

**Figure 3 micromachines-13-00940-f003:**
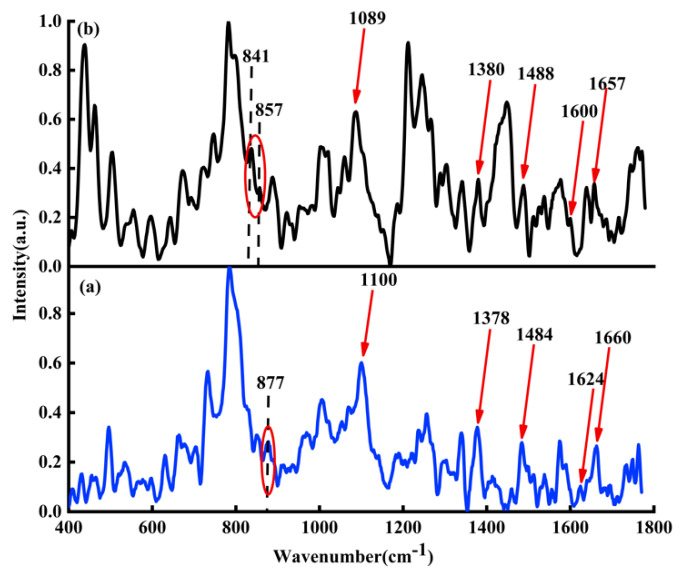
(**a**) Raman spectra of ctDNA solution (blue line) and (**b**) Raman spectrum of DOX-DNA complex (black line). The molar ratio of base-pair versus drug is two.

**Figure 4 micromachines-13-00940-f004:**
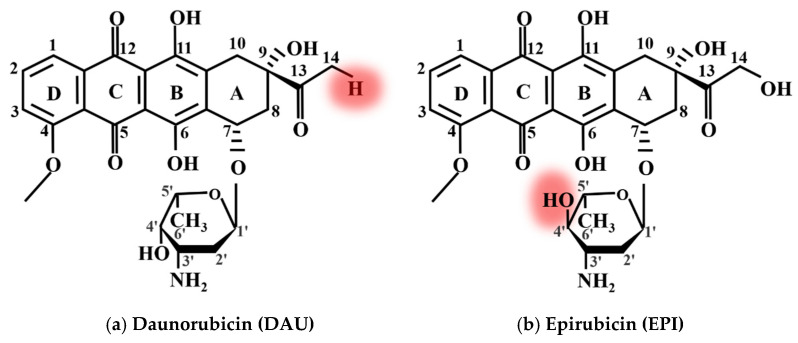
The molecular structure of (**a**) DAU and (**b**) EPI.

**Figure 5 micromachines-13-00940-f005:**
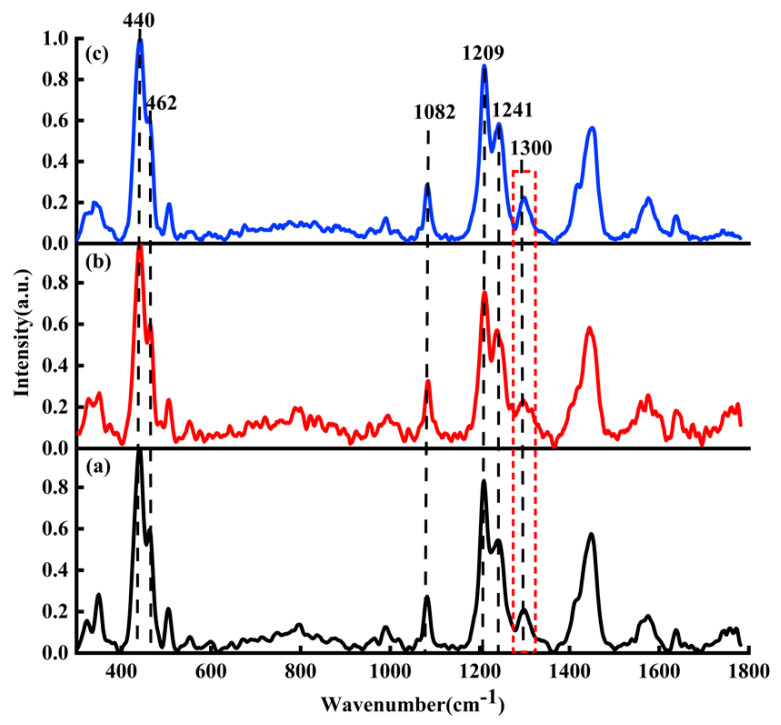
Raman spectra of (**a**) DOX (black line), (**b**) DAU (red line), and (**c**) EPI (blue line).

**Figure 6 micromachines-13-00940-f006:**
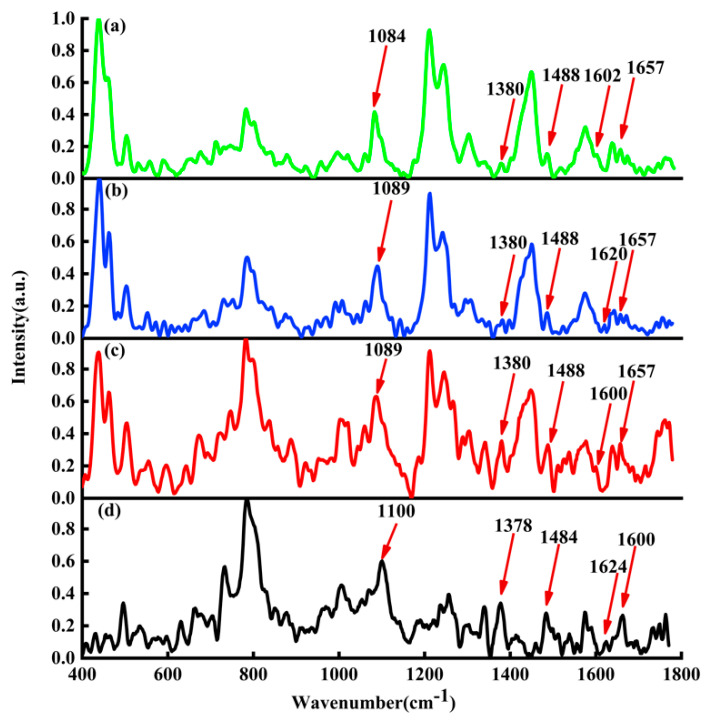
Raman spectra of (**a**) DNA (black line), (**b**) DOX-DNA complex (red line), (**c**) DAU-DNA complex (blue line), and (**d**) EPI-DNA complex (green line).

**Figure 7 micromachines-13-00940-f007:**
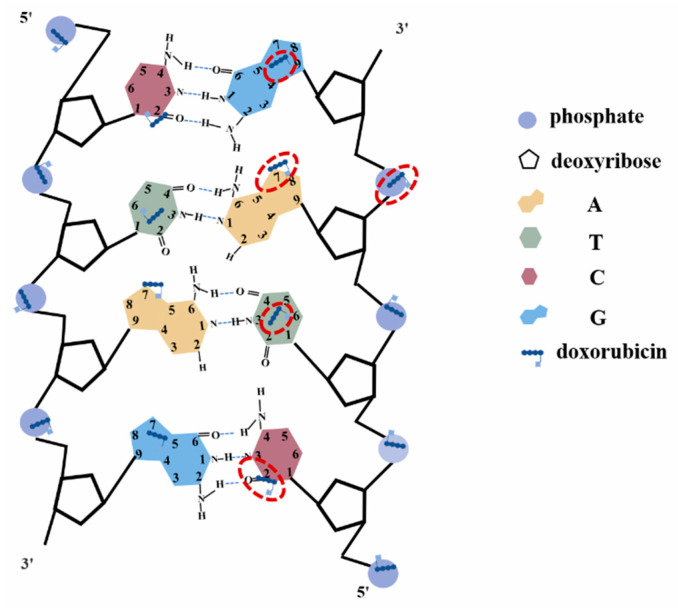
The diagram of the interaction sites of DOX and DNA.

**Figure 8 micromachines-13-00940-f008:**
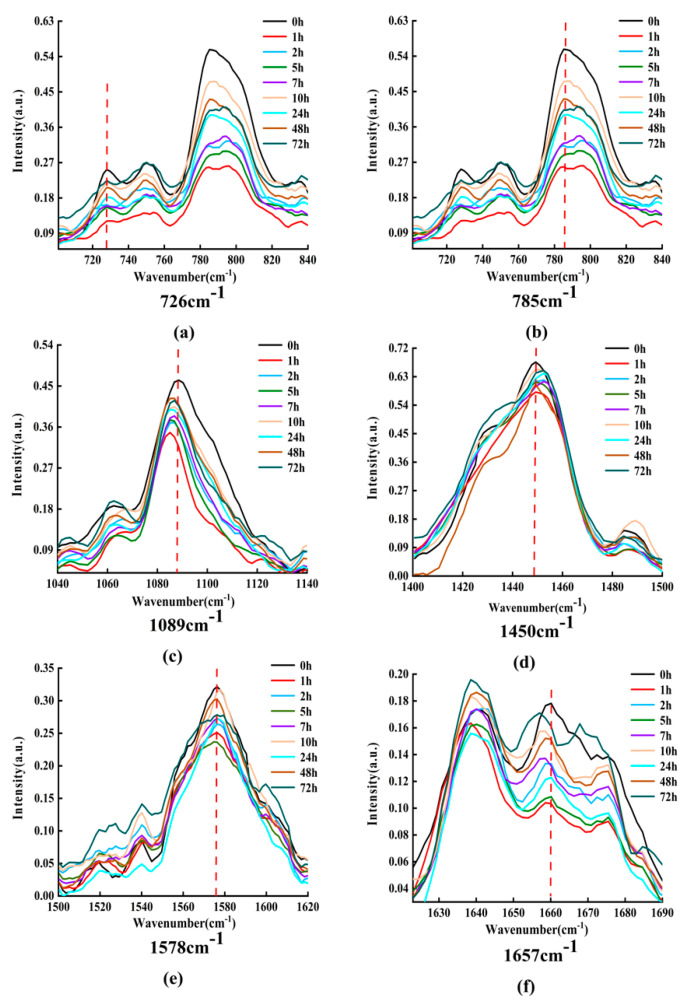
Raman spectra of DOX interacting with DNA at different times. (**a**–**f**) Raman spectrum of the band at 726, 785, 1089, 1450, 1578, and 1657${\;\mathrm{cm}}^{-1}$, respectively.

**Figure 9 micromachines-13-00940-f009:**
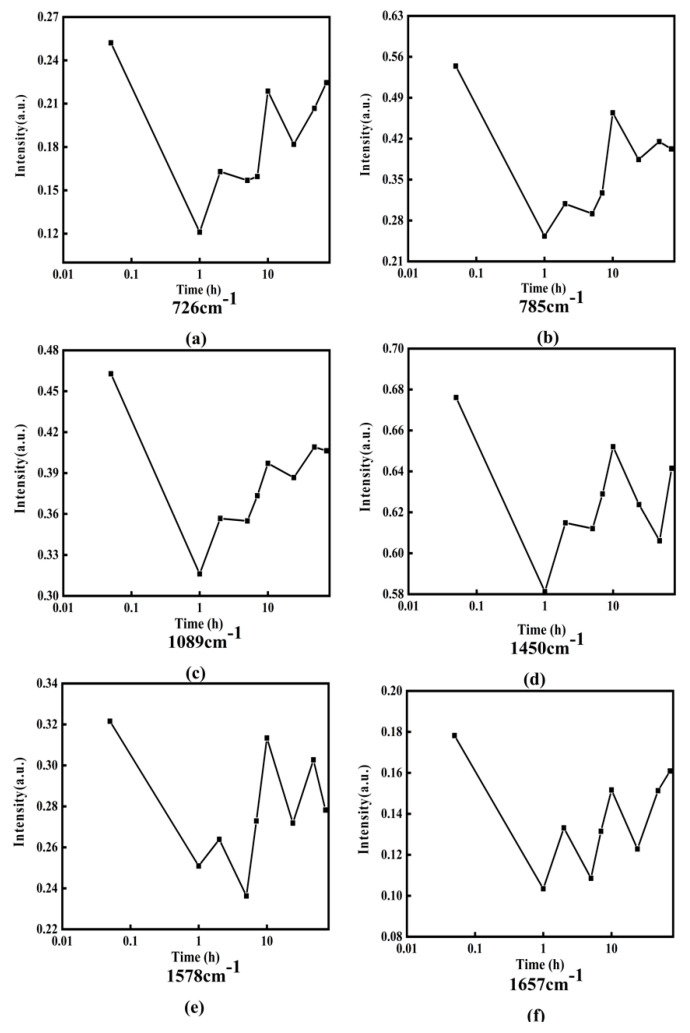
Line charts of Raman intensity change with incubation time at various wave numbers $({\mathrm{cm}}^{-1}$): (**a**) 726, (**b**) 785, (**c**) 1089, (**d**)1450, (**e**) 1578, and (**f**) 1657.

**Figure 10 micromachines-13-00940-f010:**
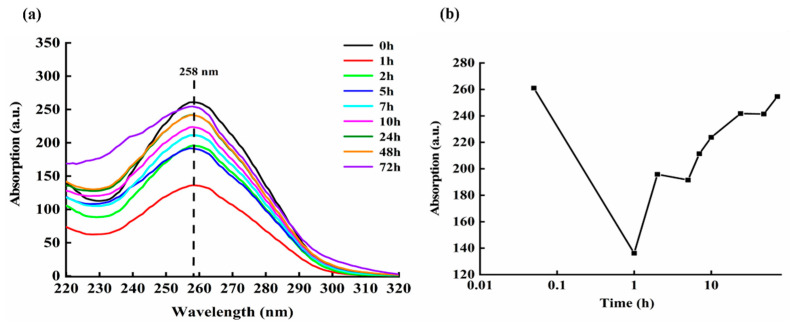
(**a**) Absorption spectra of the DOX-DNA complex with different incubating times; (**b**) absorption peak values of DOX-DNA complex around 258 nm at various incubation times.

**Figure 11 micromachines-13-00940-f011:**
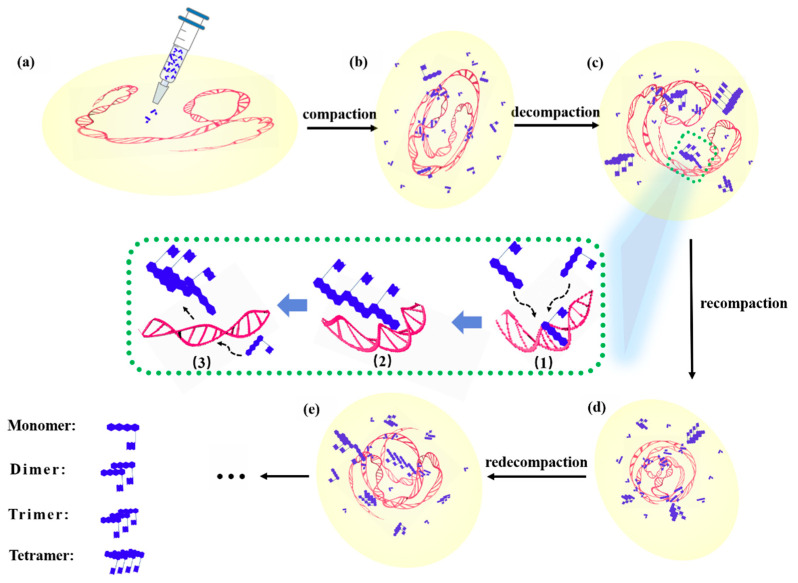
Schematic diagram of the DNA configuration changing process interacted by DOX. (**a**–**e**) Graphics of DOX interaction with DNA at different times. The insets in the middle indicate the detailed process of forming a DOX trimer. (1) Free two DOX molecules are attracted to the DOX-DNA complex; (2) A trimer is formed; (3) The trimer leaves the DOX-DNA complex and other doxorubicin molecules will interact with DNA.

**Table 1 micromachines-13-00940-t001:** Raman spectrum (wave number,${\;\mathrm{cm}}^{-1}$ ) and attribution of ctDNA and DOX interacting with ctDNA [20,30,31,32,33,34,35,36].

$\mathrm{DNA}/cm^{-1}$	$\mathrm{DOX}-\mathrm{DNA}/cm^{-1}$	Assignment	Ref.
733		Adenine (A)	[33,34,36]
785		Cytosine (C)	[33,34,36]
	841	B-DNA	[30,31,32,33]
	857	C-DNA	[30,31,32,33]
877		C-DNA	[30,31,32,33]
1006		deoxyribose	[33,34,36]
1100	1089	symmetric stretching vibration of O=P=O in ${\mathrm{PO}}_{2}{}^{-}$	[33,34,36]
1378	1380	Thymine (T)	[20,35,36]
1484	1488	Guanine (G)	[20,35,36]
1624	1600	N7C5 + C8N7 vibration of adenine (A)	[20,35,36]
1660	1657	C2 = O stretching vibration of cytosine(C)	[33,34,36]

## Data Availability

Not applicable.

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
