# Peer review of "The Mechanism of Dynamic Interaction between Doxorubicin and Calf Thymus DNA at the Single-Molecule Level Based on Confocal Raman Spectroscopy"

_micromachines, 2022, doi:10.3390/mi13060940_

Round 1
Reviewer 1 Report
A good article, a competent introduction structure, and the role of scientific research among the works on the study of DNA and DOX using spectral methods is clearly expressed. But there are a number of questions-comments on the research methods and the obtained results.
1. (lines 80-84) Why was confocal Raman used as the research method? What information is cut off by the optical circuit? Why is conventional microscopic Raman microscopy not suitable for examining dried DNA solutions on a gold substrate?
2. (lines 104-106) "Raman spectra were acquired by an Alpha 500R Confocal Raman microscopy system (Alpha 500R, WITec GmbH, Germany). Briefly, a fiber-coupled 785 nm semiconductor laser was collimated into a 20× objective lens” – what was the power of the laser? What was the spot size using a 20× objective, what was the power density, respectively, was heat removed from the sample? Were there any thermal damage to the test solutions?
3. (lines 130-133) Why are there no bands 733, 785 and 1006 cm-1 in table 1?
4. "Table 1. Raman spectrum (wave number, ??−1 ) and attribution of ctDNA and DOX interacting with ctDNA at 785nm" - what does 785 nm mean at the end? Incorrect table name or remove or supplement the name.
5. (lines 148-149 and Figure 2) Why is there no mention of the bands around 1300 and 1450 cm-1 (one of the most intense bands)? Their interpretation is missing in the text. There is no analysis of them in the study of DOX, DAU and EPI and in their interaction with DNA.
6. (lines 173-174 and Figure 3) 1600 and 1624 cm-1 bands, the intensity of these bands is low, below the noise level in the presented spectra. These bands are then used to further analyze DNA interactions (Figure 6). How reliable are these results?
7. Figure 5. What is its meaning? There is a lack of analysis of the presented spectra. How was the spectrum normalized? Are there any differences in these spectra?
8. (lines 228-232 and Figure 9) An interesting assumption has been made that describes the change in the intensities of the Raman bands. But the question arises, how many times was the dependence of the band intensities on time studied, only once or was there a series of experiments? That is, if there was a single measurement of the spectra, then such a dependence may turn out to be random, and the periodic drop and rise in intensities will lie within the confidence interval.
9. (lines 268-270 and figure 10) The same as in the previous paragraph, only in figure 10 there are no periodic changes at all after 1 hour.
10. Figure 11 Schematic diagram of the DNA configuration changing process interacted by DOX - is this an assumption about the configuration or is there any reference method that confirms this fact or at least can confirm it?
11. Comments on the graphs: Figure 1 - intensity value 1.2 (why is the value greater than 1 on the y-axis?). Perhaps it would be clearer to superimpose the blue spectrum on the black spectrum, so that it is in one picture (Figure 1.b); figure 2.b - intensity values ​​greater than 1; figure 3 - sequence a and b, it is possible to make one picture out of them; picture 5 - the intensity value is greater than 1, again, it can be combined into one picture; figure 6 - the intensity value is greater than 1; figure 8 - small figures, values ​​on the axes are not visible.
Author Response
Thank you very much for your warm and patient work, we appreciate your constructive and helpful comments and suggestions. Those comments are all valuable and very helpful for revising and improving our paper. We have carefully studied all the comments and have addressed all of them point to point.

Reviewer 2 Report
The manuscript “The mechanism of dynamic interaction between doxorubicin 2 and calf thymus DNA at the single-molecule level based on 3 Confocal Raman spectroscopy” examines binding sites and dynamics of the interaction between doxorubicin and DNA. The work is very interesting and well structured, but some issues have to be addressed before I can recommend publication:
1. What are the reciprocal concentrations of DOX and DNA used in the experiments? This information is particularly interesting to understand whether under the adopted experimental conditions it is possible to observe DOX-DOX multimers since a limit of CDOX/CDNA>0.3 for their formation is given in the text (p9, line 248).
2. Experiments seem to show that DOX and EPI have the same action: do the authors know if they actually have the same effectiveness? Is DAU less active?
3. P7, line 210: How is the appropriateness determined? Do the authors expect differences related to the change in concentration of species in the solution? Please comment.
4. P8, lines 217-228: What happens to T and G signals during incubation? it has been shown previously that DOX also interacts with these bases, at least after 24h; however, is the process dynamic? Are interactions with these bases established only after a certain period of incubation?
5. P9, lines 233-236: can the authors explain further? Is the reduction in signaling related to the shifts induced by DOX bound? Did you check or do you expect that increasing the amount of DOX relative to the amount of DNA the trends are the same, but the shifts more pronounced?
6. Are there typical signals indicating the formation of DOX-DOX multimers? Can they be recognized in the spectra considered i.e., can a variation corresponding to the fluctuation in the intensity of DOX-DNA signals be observed?
Minor
a) Could the authors indicate how many spectra were collected and averaged per sample in the materials and methods section?
b) Figures 1b and 2b: Why is the DNA spectrum shown for both solution and powder and that of DOX only in solution?
c) P4, lines 134-135: I suggest clarifying that 1100 cm-1 signal originates from the symmetric dioxy-stretch of the phosphate backbone so that the PO2- structure to indicate the phosphate group is not ambiguous
d) P5, lines 161-162 lines: the sentence “the characteristic peak 161 at 877 cm−1 in the Raman spectrum of ctDNA solution appears” seems to suggest that this signal was not present in DNA original spectrum, please rephrase to avoid confusion.
e) I suggest moving the diagram of DOX-DNA interactions proposed in fig7 to fig3.
f) Fig 5: it would be useful to highlight the differences between the spectra mentioned in the text
g) Fig8: Please, improve the image quality
Author Response
Thank you very much for your warm and patient work, we appreciate your constructive and helpful comments and suggestions. Those comments are all valuable and very helpful for revising and improving our paper. We have carefully studied all the comments and have addressed all of them point to point. In the revision manuscript, some new illustrations and figures were added according to the supplementing experiments. However, these changes did not influence the content and framework of the paper. As can be seen from the resubmission where the changing of texts had been traced by “Track Changes”, a significant revision has been made on the manuscript.

Round 2
Reviewer 1 Report
Almost all comments were worked out by the authors of the article. But there are a few details that I want to focus on:
1) previous comment 5 and author's answer -
Comment:
5. (lines 148-149 and Figure 2) Why is there no mention of the bands around 1300 and 1450 cm-1 (one of the most intense bands)? Their interpretation is missing in the text. There is no analysis of them in the study of DOX, DAU and EPI and in their interaction with DNA.
Reply:
This is a good question.
The intensities of bands around 1300 and 1450 cm-1are pronounced, however, they are not the crucial criterion to identify whether the solute is DOX (or DAU/EPI). Therefore, the bands around 1300 and 1450 cm-1 are generally not used for special analysis.
No interpretation of these two bands in the text and in the answer. "however, they are not the crucial criterion to identify whether the solute is DOX (or DAU/EPI). Therefore, the bands around 1300 and 1450 cm-1 are generally not used for special analysis." after interpretation bands, I think you should add this text in the article.
2) previous comment 7 and author's answer -
Comment:
7. Figure 5. What is its meaning? There is a lack of analysis of the presented spectra. How was the spectrum normalized? Are there any differences in these spectra?
Reply:
Thanks, and this is also a good question.
1) Figure 5 (a)-(c) are The Raman spectra of DOX, DAU and EPI, respectively.
2) In this study, because the DOX, DAU and EPI have similar structures, DAU and EPI were mainly used to verify the binding sites between DOX and DNA, that is, the adenine, thymine, cytosine, guanine and phosphate.
3) Figure 5 was only used to illustrate the structural similarity of DOX, DAU and EPI. While, their differences are not the focus of this study. Therefore, there are no in-depth analyses about the Raman spectra of the different drugs in Figure 5.
4) In the process of normalizing the ordinate of the Raman spectrum, the characteristic peaks corresponding to the maximum value of all spectra are consistent, so different spectra are comparable.
Paragraphs 2, 3 and 4 (green color) should be included in the text of the work in a modified form, so that it is clear why Figure 5 is shown.
3) previous comment 8 and author's answer -
Comment:
8. (lines 228-232 and Figure 9) An interesting assumption has been made that describes the change in the intensities of the Raman bands. But the question arises, how many times was the dependence of the band intensities on time studied, only once or was there a series of experiments? That is, if there was a single measurement of the spectra, then such a dependence may turn out to be random, and the periodic drop and rise in intensities will lie within the confidence interval.
Reply:
Thanks, this question is very meaningful.
During our experiments, in order to ensure the reliability of the measurement data, the experiment was repeated 6 times under each different working condition, and then, the average value was taken as the final result. In fact, each of the six experimental results showed similar variation.
Therefore, the periodic drop and rise of spectral intensity is not random.
The paragraph (green colour) should be included in the text of the work (my opnion).
4) previous comment 10 and author's answer -
Comment:
10. Figure 11 Schematic diagram of the DNA configuration changing process interacted by DOX - is this an assumption about the configuration or is there any reference method that confirms this fact or at least can confirm it?
Reply:
Thanks, this is a good question.
Figure 11 is a mechanism model of interactions between DOX and DNA, which is firstly proposed by us, based on our experimental results and those of other peer research groups. About this model, we are doing further investigations, e.g., by using Dynamic Light Scattering (DLS), Atomic Force Microscopy (AFM) and so on, to determine the magnitude and type of force at the action site, the influence of DOX on the size and morphology of DNA molecules.
The paragraph (green colour) should be included in the text of the work (my opnion).
Author Response
Thanks for your suggestions on revising the manuscript, and we have made modification accordingly. Hope the current version can be acceptable.
